# A prospective cohort study to describe the morphology of buboes in patients with bubonic plague using ultrasound imaging

Josephine Bourner[1]*, Reena Dwivedi[2], Rindra Vatosoa Randremanana[3],
Lisy Hanitra Razananaivo[3], Elise Pesonel[1], Esteban Garcia-Gallo[1], Elizabeth Joekes[2],
Théodora Mayouya-Gamana[3], Elisée Randriambolamanana Fanomezantsoa[3],
Aly Ny Aina Minoarisoa[3], Ezra Rajoeliarivelo[3], Mamy Gabriel Randriamanantsoa[4],
Alex Paddy Abdel Salam[1], Amanda Rojek[1], Tojo Rémi Rafaralahivoavy[3],
Mihaja Raberahona[4], Piero Olliaro[1]

**1** ISARIC, Pandemic Sciences Institute, University of Oxford, Oxford, United Kingdom, **2** Worldwide Radiology, Liverpool, United Kingdom, **3** Institut Pasteur de Madagascar, Antananarivo, Madagascar, **4** Centre Hospitalier Universitaire Joseph Raseta Befelatanana, Antananarivo, Madagascar

\* josephine.bourner@ndm.ox.ac.uk

## Abstract

### Background

Bubonic plague, caused by *Yersinia pestis*, is characterised by painful, enlarged lymph nodes ("buboes"). Despite centuries of clinical recognition, bubo morphology has been described only through observation and palpation. This study aimed to characterise the sonographic features of buboes over time in confirmed bubonic plague and evaluate the validity of digital calliper measurements compared to ultrasound.

### Methods/principle findings

We conducted a prospective cohort study at three rural health centres in Madagascar between January and March 2024. Participants with suspected bubonic plague underwent ultrasound imaging and digital calliper measurements of enlarged lymph nodes at inclusion (D1), and follow-up on D4 and D11. Bubo size and morphology were assessed by clinicians – who received targeted ultrasound training for the study – using portable ultrasound devices, with expert radiologist oversight. Neither clinicians or radiologists were blind to clinical information or outcomes. Final diagnoses were retrospectively assigned using WHO criteria and national laboratory results. Of 16 enrolled participants, 12 were confirmed plague cases. Most buboes exhibited normal morphology on D1, with limited change over time despite clinical improvement. No association was found between bubo size or morphology and clinical status. Digital calliper measurements differed substantially from ultrasound. Study sonographers achieved good agreement with radiologists on bubo size, but lower agreement on structural features.

**Data availability statement:** Aggregated data is available and openly accessible via the following link: https://projects.vertex.isaric.org/?param=Bubo_Images. The protocol is available on the ISARIC website: https://isaric.org.

**Funding:** This work was supported by the Moh Family Foundation (PSR00010 to P.H.) - https://www.themohfoundation.org. The funders had no role in study design, data collection and analysis, decision to publish, or preparation of the manuscript. None of the authors received a salary from the funder.

**Competing interests:** The authors have declared that no competing interests exist.

## Conclusions/significance

Bubo morphology and size do not appear to correlate with clinical status, challenging their use as indicators of treatment response. Digital callipers introduce significant measurement error. Newly trained clinicians can perform size measurements reliably, but further training is needed for accurate sonographic characterisation.

### Author summary

Bubonic plague, caused by *Yersinia pestis*, presents with painful, enlarged lymph nodes known as buboes. While their clinical significance has been recognised for centuries, existing descriptions rely on clinical evaluation alone. A recent systematic review reported buboes in 96% of confirmed cases, but provided little detail on their evolution and morphology. Imaging has been reported in a small number of case reports, usually at a single timepoint, and without radiological characterisation. This is the first study to describe the sonographic features of buboes in a cohort of confirmed bubonic plague cases using portable ultrasound. It provides longitudinal ultrasound data from a rural endemic setting in Madagascar and compares clinical and ultrasound bubo size measurements. We found that bubo morphology and size remained largely unchanged during treatment, despite clinical improvement, challenging the assumption that reduction in bubo size reflects clinical recovery. Calliper measurements also consistently overestimating size, and while newly trained clinicians showed strong agreement with radiologists for size, their interpretation of internal structures was less consistent. Overall, our findings indicate that bubo size and morphology are poor indicators of treatment response and may be unreliable endpoints for clinical trial or patient monitoring.

## Background

Bubonic plague, caused by gram-negative bacteria *Yersinia pestis*, has been responsible for three pandemics in modern history, killing approximately one third of the European population in the 14th century. [1] Evidence of *Y. pestis* infection can be traced back more than 5000 years. [2] Despite humanity's long history with plague, little is known about its characteristic feature, enlarged lymph nodes referred to as "buboes".

The number of plague cases reported globally each year varies from approximately 250–700. [3] However, 80% of the global burden lies in Madagascar. [3] Plague has existed in Madagascar for more than 100 years since 1898 [4], initially occurring primarily in coastal areas before spreading to the central highlands, where it is endemic to this day. [5]

There are three main clinical forms of plague – bubonic, pneumonic and septicaemic – the most common being bubonic plague. Transmission to humans usually

occurs by the bite of an infected flea or contact with infected bodily fluid or tissues. [6] In addition to the presence of a bubo, bubonic plague is typically characterised by the rapid onset of fever, headache, altered mental status and chills. [7] The Case Fatality Ratio (CFR) is around 15%. [7]

While academic literature has frequently reported the presence of a bubo in patients diagnosed with bubonic plague – reported in 96% of cases – few other details about its evolution and morphology have been described. [7] At present, our understanding is limited to observation and palpation, which have remained unchanged for centuries. This is compounded by limitations in access to diagnostic imaging in remote and rural communities who are most affected by plague, particularly in Madagascar. We therefore have a limited idea of how the structure of a lymph node and the surrounding tissues change with infection and how they evolve during treatment. To date, ultrasound imaging of the bubo has been conducted in only one study, a 2018 case report of a patient with bubonic plague in the United States. [8] Imaging of enlarged inguinal and suprapubic lymph nodes was conducted at two timepoints and demonstrated increased presence inguinal lymph nodes on repeat imaging. [8] The case report did not discuss the sonographic characteristics of the bubo, nor its evolution in detail, and no further ultrasound imaging studies of buboes have taken place since. Computed tomography of the abdomen was used in a separate case study in 1987. [9]

In addition, reduction in bubo size has been used in two clinical trials evaluating treatments for bubonic plague. [10,11] This is however in the absence of consensus on clinically relevant outcome measures for plague, evidence to suggest that bubo size has any relationship with the clinical status of the patient and the validation of measurement tools used to assess the bubo. One of the trials used a digital calliper to measure buboes [10] – which has been demonstrated to carry a high risk of measurement error [12] – and the other trial did not specify the measurement tool used [11].

Given the uncertainties around the role of the bubo and as bubo size has been used as an indicator of treatment response, the primary objective of this study was to explore the relationship between the size and morphology of a plague bubo and clinical status. Understanding this relationship could enhance both plague management and the definition of clinically meaningful clinical trial endpoints. A secondary objective of this study was to evaluate the validity of using a digital calliper to measure plague buboes by comparing calliper and ultrasound measurements.

## Methods

### Ethics statement

This study was approved by Ethics Committee for Biomedical Research of Madagascar (CERBM, approval number: IORG0000851) and the University of Oxford Tropical Research Ethics Committee (OxTREC, approval number: 565-21). Written informed consent was obtained from all study participants before any study procedures took place.

This study was a multi-site prospective cohort study of patients with enlarged lymph nodes. Study assessments were conducted at inclusion (D1) and at least one follow-up visit on D11. An additional optional visit was conducted on D4, subject to local capacity.

The study was conducted at health centres in three districts participating in the IMASOY trial – Ambositra, Manandriana and Ambohimahasoa. [10]

Participant identification and recruitment used the research infrastructure of the IMASOY trial, a randomised controlled trial that was being conducted in the same region as this study. Participants were individuals of any age presenting to participating health centres with suspected bubonic plague and regional lymphadenopathy who were willing and able to provide informed consent. Exclusion criteria included cases where the enlarged lymph nodes exhibited signs of suppuration or where high pain scores made ultrasound imaging intolerable.

Following enrolment, demographic information and biological samples were collected from all participants as part of routine care for plague and tested within the national plague surveillance programme. Based on the results, participants were retrospectively classified as confirmed cases or non-cases using the WHO plague case definition. [13] All

participants in the study received treatment for plague according to the national plague guidelines, which included either an aminoglycoside (streptomycin or gentamicin) plus ciprofloxacin or ciprofloxacin alone.

Data were collected using a standardised paper Case Report Form (CRF), which was subsequently transcribed into REDCap. [14,15]

Clinical and portable ultrasound assessments were conducted on D1, D4 and D11. The clinical assessments included a physical examination, measurement of the largest enlarged lymph node (referred to as the index node) detected in each anatomical zone (axillary, cervical and inguinal) using a digital calliper, and an overall evaluation of the patient's clinical status. We measured only the index lymph node, as is standard in clinical practice and clinical trials, where the largest or most representative node is selected for follow-up over time.

Ultrasound assessments were performed on the index node using a Butterfly iQ+ probe (iQ, Butterfly Network, Inc, Guilford, CT, USA) and included evaluation of nodal shape (oval or round), capsule (distinct or blurred), hilum appearance (visible or obscured), cortex (homogenous or irregular), presence of necrosis, hypervascularity, perinodal reactivity, and coalescence. A summary of the sonographic features evaluated in the study can be seen in Fig 1. Still images and cine clips of the index node were captured on D1, D4 and D11 for subsequent QA review by two experienced radiologists. Node measurements were obtained using the Butterfly iQ+ measurement tool.

Before the study was initiated, local clinicians and research staff (henceforth referred to as "study sonographers") underwent a targeted ultrasound training programme, covering the use of ultrasound for lymph node evaluation. Two experienced radiologists provided the face-to-face training and were responsible for the oversight of the study scanning. One of the radiologists was based in Madagascar to provide more context-appropriate discussion and training opportunities for the team. The training program involved two days of classroom-based training followed by seven days of supervised scanning, during which all images collected were subject to remote quality assurance (QA). Training was delivered in Malagasy and French to the clinical health worker cohort.

All images were independently reviewed by the two radiologists, who subsequently performed a joint review of their assessments at the study's conclusion to resolve any discrepancies in their evaluations. During the QA process, the radiologists completed a short CRF to document whether their assessments differed from those of the study sonographers. In instances where discrepancies were identified, the radiologists' evaluations were used for the final analysis. Neither clinicians nor radiologists were blind to clinical status or patient outcomes.

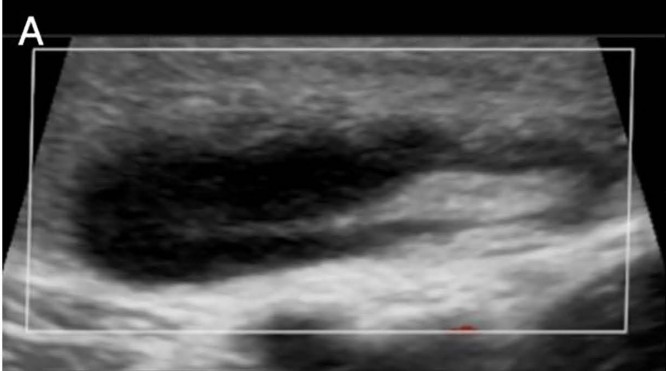 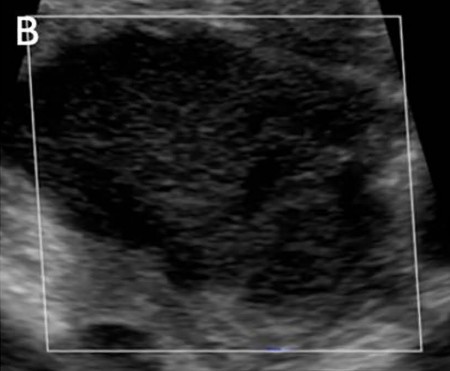

**Fig 1. Normal and abnormal lymph node features. A)** An example of a plague bubo without abnormal radiological findings. While enlarged, this lymph node exhibits relatively normal features including a well-defined, oval-shaped capsule with a distinct echogenic hilum and normal surrounding perinodal tissue. **B)** Anomalous findings on ultrasound indicative radiological progression would include blurring to the edges of the capsule (which may be indicative of oedema), a circular-shaped capsule, a decrease in the size of the hilum or a total absence of the hilum.

## Analysis

**The analysis was conducted using R v3.6.0.** Participants were retrospectively categorised as "plague cases" or "non-cases" according to the WHO plague case definition following receipt of laboratory results from samples collected as part of routine care under the national plague surveillance programme. [13] In this study, plague cases are those who received either a confirmed or probable diagnosis. Enlarged lymph nodes of plague cases are referred to as "buboes". For non-plague cases, they are referred to as "enlarged lymph nodes".

Clinical and sonographic characteristics of interest were pre-defined on the CRF. The number and proportion of participants with each characteristic at baseline (D1) and D11 are presented in a stacked pyramid chart.

A short report comparing the clinical and sonographic evolution of two clinical cases is described to highlight findings in the relationship between clinical status and radiological progression. Both cases presented are confirmed cases of plague.

Measurement validity is evaluated by comparing measurements taken by a digital calliper and those taken by ultrasound, which is used as the reference size. Bubo size is the sum of the long axis and short axis. The absolute and percentage difference between each calliper and ultrasound measurement was calculated in order to generate a mean difference overall and per zone (axillary, cervical and inguinal).

To assess agreement in ultrasound evaluations between study sonographers and radiologists, we report the absolute and percentage differences between measurements. The proportion of radiological features correctly identified by the study sonographers in comparison to the radiologists is represented in a heat map.

## Data visualization

Data visualizations were performed in ISARIC VERTEX [16], a web-based application that generates tables and visualizations to support key research questions. VERTEX is part of the ISARIC Clinical Epidemiology Platform, developed to standardize and streamline data collection and analysis in outbreak-related clinical research. A public implementation of VERTEX for this study is available at: https://projects.vertex.isaric.org/?param=Bubo_Images.

## Results

Between January and March 2024, 16 participants were enrolled in the study, of which 12 were confirmed cases of plague and four were non-cases (Table 1). An equal number of male and female participants were enrolled in both the plague and non-case groups. The median age of plague cases was 14.5 years (range: 6–50 years) and 10.5 years (range: 3–23 years) for non-plague cases. Time between symptom onset and enrolment in both groups was one day (range: 1–2 days).

Alternative diagnoses for non-plague cases are not known due to limited laboratory testing capacity at the sites involved in the study.

## Clinical characteristics

At baseline, all participants presented with a single enlarged lymph node. Among confirmed plague cases, seven inguinal and three axillary buboes were observed, compared to two enlarged cervical lymph nodes and two enlarged inguinal

**Table 1. Demographic characteristics.**

|  | All | Plague cases | Non-cases |
|---|---|---|---|
| Total number of cases enrolled | 16 | 12 | 4 |
| Male: female | 8: 8 | 6: 6 | 2: 2 |
| Age (years), median [range] | 13.5 [3 − 50] | 14.5 [6 − 50] | 10.5 [3 − 23] |
| Duration of symptoms before enrolment (days), median [IQR] | 1 [1 − 2] | 1 [1 − 2] | 1 [1 − 2] |

lymph nodes in non-cases. Most were visible (83%) and hard (58%), with few buboes exhibiting erythema (17%) or oedema (33%) (Fig 2). The median pain score of plague buboes at baseline was 6/10. All participants presented with mild plague symptoms, except one participant who presented with severe symptoms.

The majority of plague buboes exhibited normal structural characteristics (Fig 1) on ultrasound. More than half of the buboes were oval-shaped (67%), had a distinct capsule (56%) and homogeneous cortex (56%), and just under half had a distinct hilum (44%). Few buboes exhibited severe structural anomalies such as necrosis (11%), hypervascularity (22%) and coalescence (11%) (Fig 3).

By D11, all participants had completed treatment and showed signs of clinical improvement with mild persistent symptoms of plague, including the participant who initially presented in a severe condition. All buboes remained detectable by ultrasound in all plague cases, with two additional buboes detected in a single case. Few changes in clinical and sonographic characteristics were observed (Figs 2 and 3). The median pain score decreased from 6 to 3/10 (range: 0–6).

There were no detectable enlarged lymph nodes in non-cases on D11.

Bubo size did not substantially change over time for plague cases (Fig 4). The median size measured by ultrasound was 30.8mm at baseline and 29.4mm at D11. The median absolute difference between D1 and D11 size was 3.3mm (range: 0.8 -- 10.5), representing a median percentage difference of 10%.

## Case reports

The clinical and sonographic progression of two cases included in the study is presented below to illustrate the lack of observed relationship between clinical status and sonographic features of the bubo.

Case 1 was a 21-year-old male participant, who was evaluated as being in a clinically severe state at the point of admission. On palpation, a left axillary bubo was detected which was hard with surrounding oedema and painful (pain score: 8/10).

On D1, the bubo demonstrated relatively normal morphology with a defined capsule and preserved hilum, and measured 28.8mm (Fig 5). No necrosis or hypervascularity was detected.

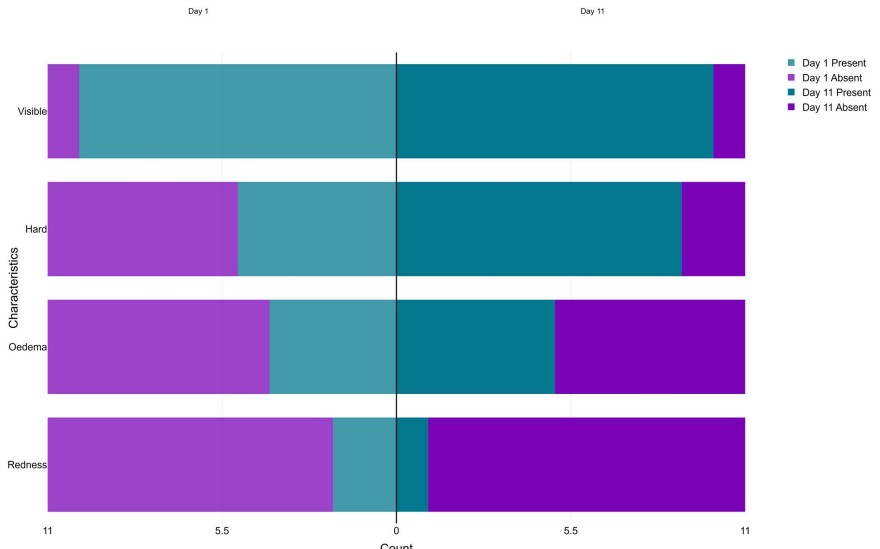

**Fig 2. Clinical characteristics of plague buboes at D1 and D11.**

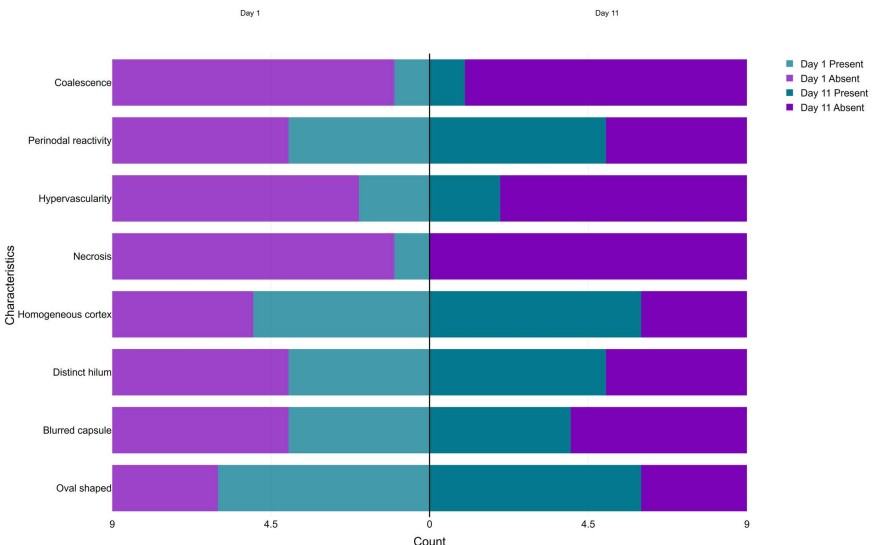

**Fig 3. Sonographic characteristics of plague buboes at D1 and D11.**

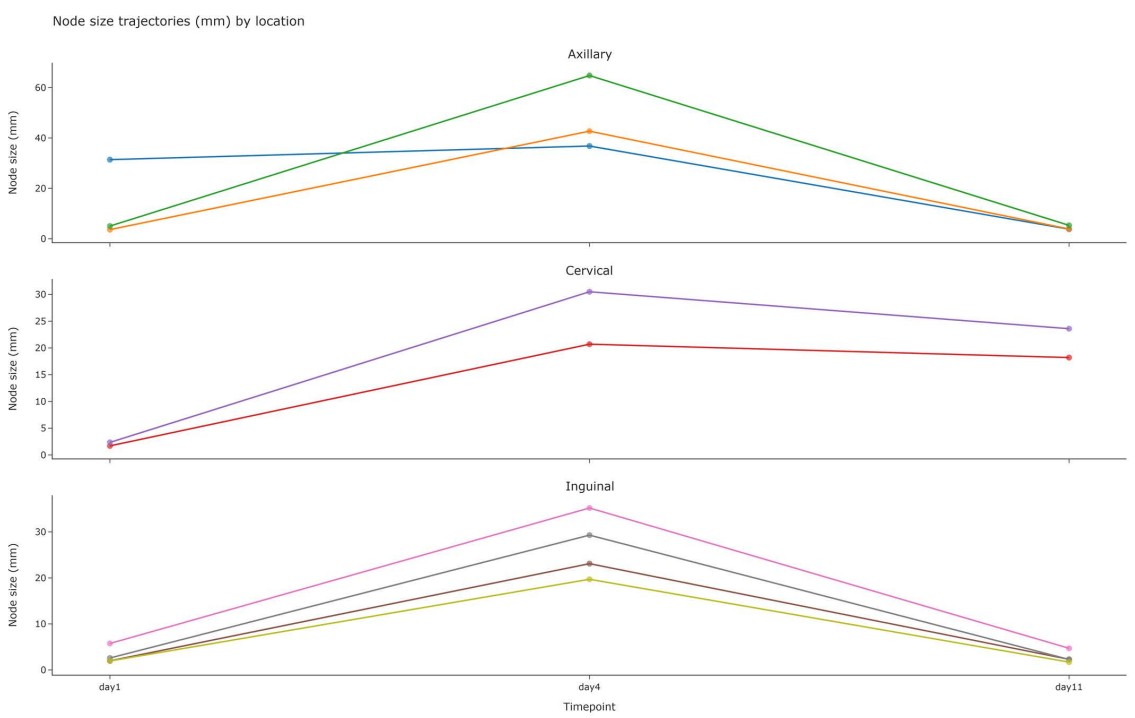

**Fig 4. Scatterplot of bubo sizes per participant on D1, D4 and D11.**

On D4, no changes to the clinical characteristics of the bubo were observed – although the patient reported a lower pain score of 6/10. The sonographic features deteriorated slightly to that observed on D1 with the nodal shape becoming more lobulated and some loss of hilarious delineation. Interruption to the capsule was also observed indicating progressive perinodal reaction and oedema. The size of the bubo increased by 7.5mm to 35.3mm.

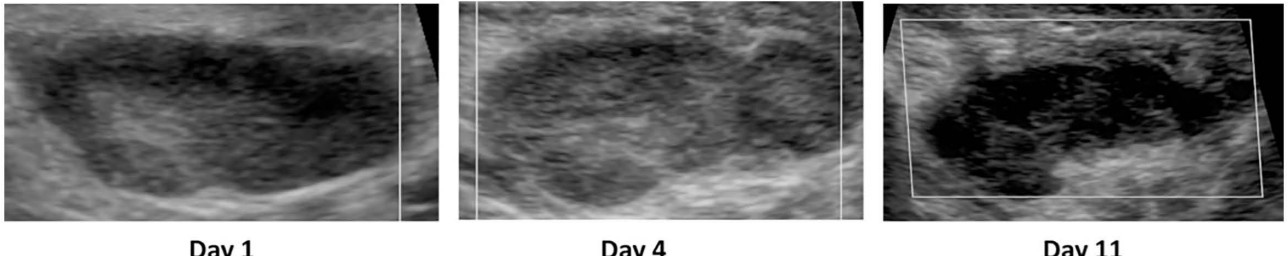

**Fig 5. Ultrasound images taken on Day 1, Day 4 and Day 11 for case report #1.**

By D11, the patient was reported to have improved clinically and only mild plague symptoms remained. While the pain score further decreased to 3/10 and no changes to the clinical characteristics of the bubo were observed, further radiological progression was present. The capsule became indistinct and irregular, the hilum was no longer visible and perinodal oedema had progressed further. The size of the bubo remained stable from the previous measurement taken on D4 at 35.4mm.

**Case 2** was a 38-year-old male participant, who presented with mild symptoms of plague at admission and a right axillary bubo, which on palpation was hard without oedema and a pain score of 8/10.

The bubo was oval-shaped with a relatively distinct capsule – although some oedema is seen in the blurred margins between the lymph node and perinodal tissue – a moderately visible hilum and homogeneous cortex (Fig 6). The bubo measured 50.3mm.

By D4, while the bubo remained hard and the pain score decreased to 6/10, oedema was noted on palpation. On ultrasound examination the bubo demonstrated interval enlargement with a rounded shape, reduced hilar definition and blurring of the capsule. The bubo increased in size from D1 by 14.5mm to 64.8mm.

By D11, the patient's clinical status had improved overall but a few mild persistent symptoms remained. The size of the bubo decreased from D4 to 53.2mm but remained larger, with loss of the normal capsule and no clear hilar echogenicity. Associated mild increase in perinodal oedema was noted.

## Measurement accuracy

Sixteen digital calliper measurements were compared to corresponding ultrasound measurements of the same enlarged lymph nodes. Substantial discrepancies were observed across all anatomical zones when comparing the two methods (Table 2).

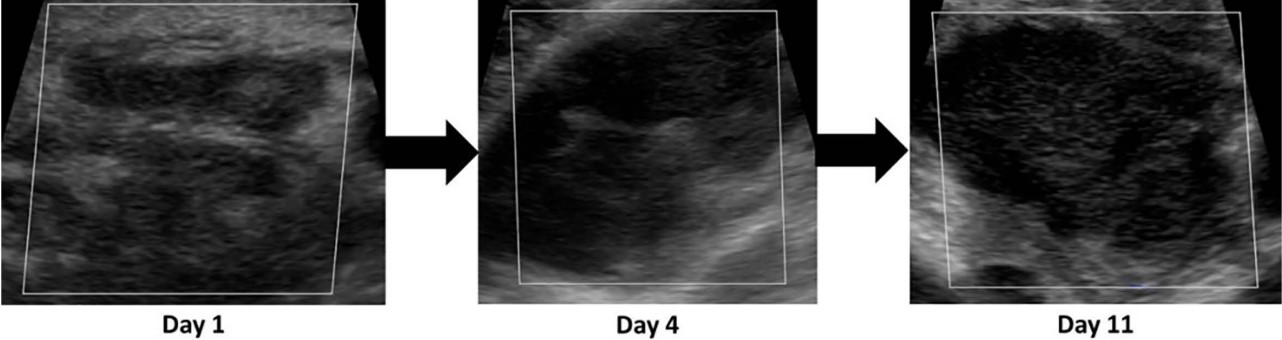

**Fig 6. Ultrasound images taken on Day 1, Day 4 and Day 11 for case report #2.**

**Table 2. Median (IQR) absolute and percentage difference between digital calliper and ultrasound measurements.**

| Location | N nodes | Absolute difference, median (IQR), mm | % difference, median (IQR) |
|---|---|---|---|
| Cervical | 1 | 2.67 (2.67-2.67) | 157.1 (157.1-157.1) |
| Inguinal | 9 | 3.15 (1.14-4.93) | 112.8 (44.1-159.9) |
| Axillary | 6 | 1.71 (0.67-2.78) | 44.53 (15.7-55.7) |

Axillary lymph nodes showed the smallest differences, with a mean absolute difference of 1.71mm (range: 0.67–2.78 mm), corresponding to a mean percentage difference of 44.5% (range: 15.7%–55.7%). Inguinal lymph nodes exhibited greater variability, with a mean absolute difference of 3.15mm (range: 1.14–4.93mm) and a mean percentage difference of 112.8% (range: 44.0%–159.9%). For cervical lymph nodes, only a single bubo was assessed using both methods, resulting in a mean absolute difference of 2.67mm and a corresponding percentage difference of 157%.

### Rater agreement

Minimal discrepancies were identified between ultrasound measurements of lymph nodes recorded by radiologists and those recorded by study technicians (Table 3). In the cervical zone, complete agreement was observed, with radiologists concurring with all three ultrasound measurements recorded by the study sonographers. In the inguinal zone, the mean absolute difference between measurements was 0.23mm (range: 0–2.58mm), corresponding to a mean percentage difference of 4.6% (range: 0–44.9%). In the axillary zone, the mean absolute difference was slightly higher at 0.375mm (range: 0–2.25mm), with a mean percentage difference of 7.5% (range: 0–44.7%).

Identification of lymph node characteristics on ultrasound however was more challenging and there was less observed agreement between the experts and non-experts (Figs 7 and 8). There was overall good agreement on the distribution of the lymph nodes, the presence of necrosis, hypervascularity and coalescence. Least agreement was observed relating to the homogeneity of the cortex. Agreement however varied by zone. Higher levels of agreement were observed in the cervical and inguinal zone, and lower levels were observed in the axillary zone.

Lower agreement was also observed on overall on the D11 scans (Fig 8) with substantially lower agreement occurring in the characterisation of the capsule and presence of necrosis.

### Discussion

This study presents findings from a small cohort of patients with suspected bubonic plague and is the first study to systematically characterise the progression of buboes from admission to the end of treatment (D11).

All confirmed plague cases presented with a detectable bubo at admission, most of which exhibited typical clinical and sonographic features of enlarged lymph nodes. Only a few cases demonstrated severe irregularities, such as hypervascularity and necrosis.

Our findings highlight the absence of an observed association between bubo size, morphology, and clinical status, which is particularly evident in the case reports. While a larger dataset would be necessary to confirm this finding, our

**Table 3. Median (IQR) absolute and percentage difference between radiologists and study sonographers.**

| Location | N nodes | Absolute difference, median (IQR), mm | % difference, median (IQR) |
|---|---|---|---|
| Cervical | 3 | 0 (0-0) | 0 (0-0) |
| Inguinal | 13 | 0.23 (0-2.58) | 4.6 (0-44.9) |
| Axillary | 6 | 0.375 (0-2.25) | 7.5 (0-44.7) |

Agreement Rate

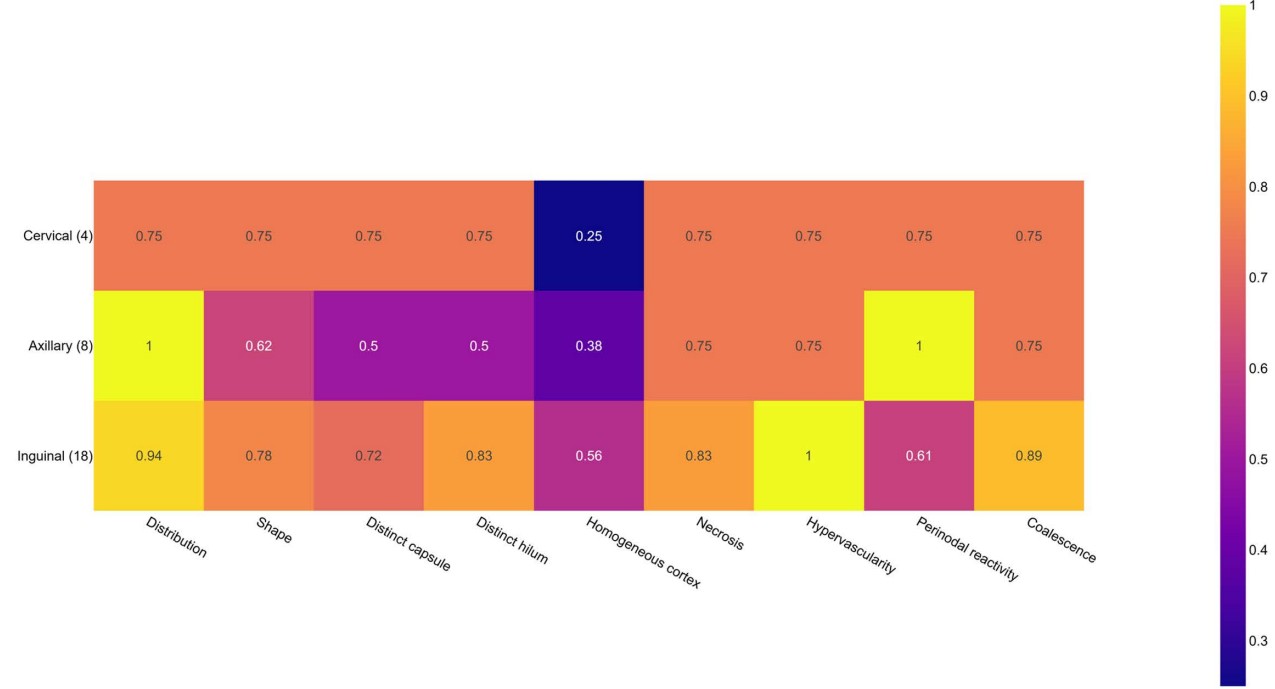

**Fig 7. Agreement between experts and non-experts by characteristic and zone.**

Agreement Rate

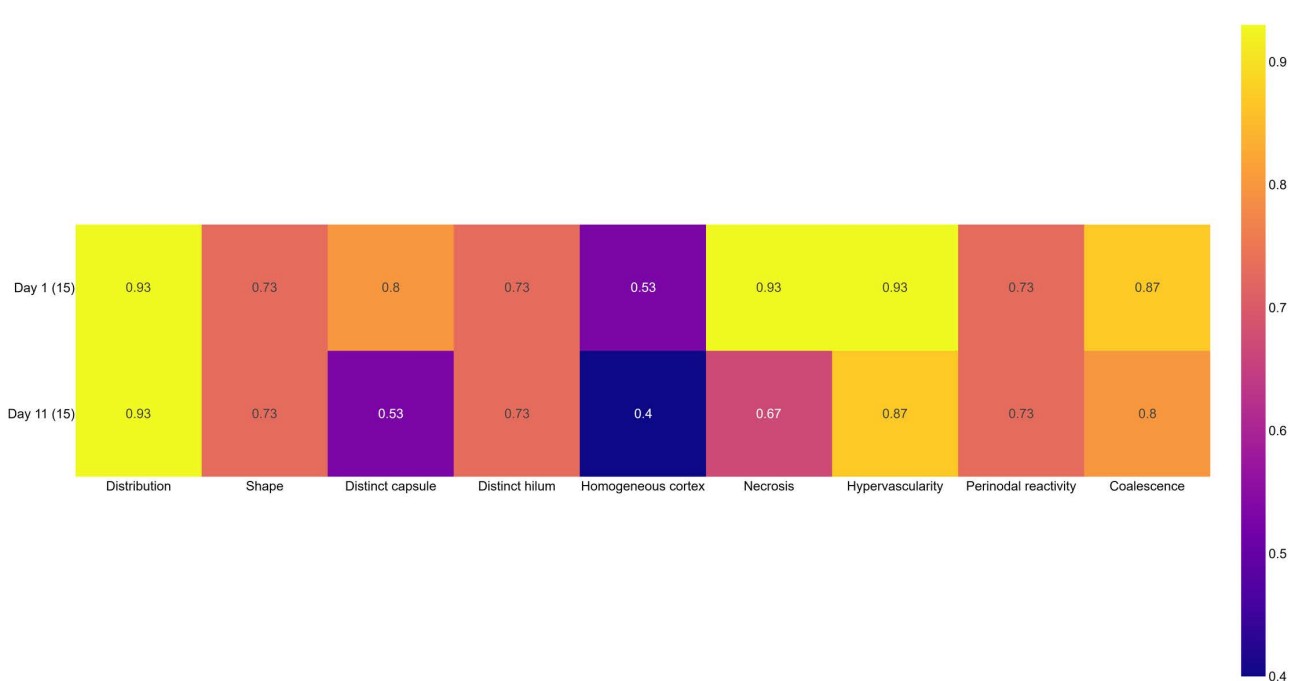

**Fig 8. Agreement between experts and non-experts by characteristic and study day.**

observations suggest that bubo size may not accurately reflect treatment efficacy and could potentially confound the interpretation of patient outcomes. This is particularly important for clinical trials, where bubo size is used to estimate treatment effects.

Our study further highlights the limitations and risks associated with using a digital calliper to measure buboes. Consistent with the findings in a separate study evaluating measurement validity and reliability of artificial buboes [12], the digital calliper measurements recorded in this study differed substantially from those measured by ultrasound, which was used as the reference standard. Measurement error was considerable across all anatomical regions assessed, with the greatest discrepancies observed in the inguinal region, where most buboes were located. In addition to the lack of association between bubo morphology and treatment response, this study therefore demonstrates that using a digital calliper to measure buboes in clinical trials may lead to incorrect classifications of patients as either having or (more likely) not having had a treatment response.

The study was conducted by clinicians and nurses with limited or no prior experience in ultrasound imaging. Training clinical staff with minimal prior sonographic experience to perform focussed and targeted ultrasound assessment for specific superficial features, such as assessment of lymph node characteristics, may contribute to improved patient management and support the wider public health system through a more sustainable approach to improving local capacity. When combined with the use of portable ultrasound devices, this approach could increase access to sonographic assessment for patients in remote or low resource settings. Additionally, it may prove valuable in managing high case burdens by facilitating more effective patient triage. While a comparison of the lymph node measurements revealed good agreement between the radiologists and the study sonographers, there was limited agreement in the identification of lymph node characteristics. This may be due to the perceived subjectivity of some of the assessments, such as the distinctness of the capsule and homogeneity of the cortex. These results suggest that while objective measures can be performed well by newly trained clinical staff, characterisation of clinical features of lymph nodes may require a longer training and supervision period with regular expert reviews.

In conclusion, our findings show that bubo size and morphology are unreliable indicators of treatment response, and that measurements by digital calliper carry substantial error. These limitations highlight the risk of misclassifying outcomes when bubo-based endpoints are used in clinical trials. Expanding access to focussed ultrasound through task shifting to appropriately trained clinical staff however may offer a more accurate and sustainable approach for evaluating therapeutic efficacy.

## Author contributions

**Conceptualization:** Josephine Bourner, Rindra Vatosoa Randremanana, Alex Paddy Abdel Salam, Mihaja Raberahona, Piero Olliaro.

**Data curation:** Josephine Bourner, Elise Pesonel.

**Formal analysis:** Josephine Bourner, Reena Dwivedi, Elise Pesonel, Esteban Garcia-Gallo.

**Investigation:** Josephine Bourner, Reena Dwivedi, Rindra Vatosoa Randremanana, Lisy Hanitra Razananaivo, Elise Pesonel, Théodora Mayouya-Gamana, Elisée Randriambolamanana Fanomezantsoa, Aly Ny Aina Minoarisoa, Ezra Rajoeliarivelo, Alex Paddy Abdel Salam, Tojo Rémi Rafaralahivoavy, Mihaja Raberahona.

**Methodology:** Josephine Bourner, Reena Dwivedi, Rindra Vatosoa Randremanana, Elise Pesonel, Alex Paddy Abdel Salam, Tojo Rémi Rafaralahivoavy, Mihaja Raberahona, Piero Olliaro.

**Project administration:** Josephine Bourner, Lisy Hanitra Razananaivo, Elise Pesonel, Théodora Mayouya-Gamana.

**Resources:** Rindra Vatosoa Randremanana, Elizabeth Joekes, Théodora Mayouya-Gamana, Mamy Gabriel Randriamanantsoa, Amanda Rojek, Mihaja Raberahona, Piero Olliaro.

**Supervision:** Rindra Vatosoa Randremanana, Alex Paddy Abdel Salam, Amanda Rojek, Tojo Rémi Rafaralahivoavy, Mihaja Raberahona, Piero Olliaro.

**Validation:** Reena Dwivedi, Elise Pesonel.

**Visualization:** Josephine Bourner, Esteban Garcia-Gallo.

**Writing – original draft:** Josephine Bourner.

**Writing – review & editing:** Josephine Bourner, Reena Dwivedi, Rindra Vatosoa Randremanana, Lisy Hanitra Razananaivo, Elise Pesonel, Théodora Mayouya-Gamana, Alex Paddy Abdel Salam, Amanda Rojek, Mihaja Raberahona, Piero Olliaro.

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
