## [Decision Letter · Decision Letter 0]

14 Oct 2025

PNTD-D-25-01601

A prospective cohort study to describe the morphology of buboes in patients with bubonic plague using ultrasound imaging

Dear Dr. Bourner,

Thank you for submitting your manuscript to PLOS Neglected Tropical Diseases. After careful consideration, we feel that it has merit but does not fully meet PLOS Neglected Tropical Diseases's publication criteria as it currently stands. Therefore, we invite you to submit a revised version of the manuscript that addresses the points raised during the review process.

Please submit your revised manuscript within 60 days Dec 13 2025 11:59PM. If you will need more time than this to complete your revisions, please reply to this message or contact the journal office at plosntds@plos.org. Please include the following items when submitting your revised manuscript:

We look forward to receiving your revised manuscript.

Kind regards,

Joseph M. Vinetz

Section Editor

Joseph Vinetz

Section Editor

Shaden Kamhawi

co-Editor-in-Chief

Paul Brindley

co-Editor-in-Chief

**Journal Requirements:**

1) Please upload all main figures as separate Figure files in .tif or .eps format. For more information about how to convert and format your figure files please see our guidelines:

2) In the online submission form, you indicated that your data will be submitted to a repository upon acceptance. We strongly recommend all authors deposit their data before acceptance, as the process can be lengthy and hold up publication timelines. Please note that, though access restrictions are acceptable now, your entire minimal dataset will need to be made freely accessible if your manuscript is accepted for publication. This policy applies to all data except where public deposition would breach compliance with the protocol approved by your research ethics board. If you are unable to adhere to our open data policy, please kindly revise your statement to explain your reasoning and we will seek the editor's input on an exemption.

3) Please amend your detailed Financial Disclosure statement. This is published with the article. It must therefore be completed in full sentences and contain the exact wording you wish to be published.

4) Please send a completed 'Competing Interests' statement, including any COIs declared by your co-authors. If you have no competing interests to declare, please state "The authors have declared that no competing interests exist". Otherwise please declare all competing interests beginning with the statement "I have read the journal's policy and the authors of this manuscript have the following competing interests"

**Reviewers' Comments:**

Reviewer's Responses to Questions

**Key Review Criteria Required for Acceptance?**

**Methods**

-Are the objectives of the study clearly articulated with a clear testable hypothesis stated?

-Is the study design appropriate to address the stated objectives?

-Is the population clearly described and appropriate for the hypothesis being tested?

-Is the sample size sufficient to ensure adequate power to address the hypothesis being tested?

-Were correct statistical analysis used to support conclusions?

-Are there concerns about ethical or regulatory requirements being met?

Reviewer #1: (No Response)

Reviewer #2: -Are the objectives of the study clearly articulated with a clear testable hypothesis stated?

Yes

-Is the study design appropriate to address the stated objectives?

Yes

-Is the population clearly described and appropriate for the hypothesis being tested?

Yes

-Is the sample size sufficient to ensure adequate power to address the hypothesis being tested?

Not relevant in this study.

-Were correct statistical analysis used to support conclusions?

Not relevant

-Are there concerns about ethical or regulatory requirements being met?

Yes.

**Results**

-Does the analysis presented match the analysis plan?

-Are the results clearly and completely presented?

-Are the figures (Tables, Images) of sufficient quality for clarity?

Reviewer #1: (No Response)

Reviewer #2: -Does the analysis presented match the analysis plan?

Yes.

-Are the results clearly and completely presented?

Yes.

-Are the figures (Tables, Images) of sufficient quality for clarity?

Needs improvement.

vide infra.

**Conclusions**

-Are the conclusions supported by the data presented?

-Are the limitations of analysis clearly described?

-Do the authors discuss how these data can be helpful to advance our understanding of the topic under study?

-Is public health relevance addressed?

Reviewer #1: (No Response)

Reviewer #2: -Are the conclusions supported by the data presented?

Yes.

-Are the limitations of analysis clearly described?

Yes.

-Do the authors discuss how these data can be helpful to advance our understanding of the topic under study?

Yes.

-Is public health relevance addressed?

Not relevant.

**Editorial and Data Presentation Modifications?**

Reviewer #1: (No Response)

Reviewer #2: (No Response)

**Summary and General Comments**

Reviewer #1: A topic of prime interest, desserved by a poorly designed, poorly reported study so that

reader (perhas even authors) learn nothing after reading pages of report.

I propose :

1. To shorten, especially the Discussion ; as well as the « two cases » report.

2. To focus the goal : It could have been the neosemiology of plague, based on

ultrasound imaging.

3. If so :

a. Clearly report the form of X parameters used to remotely analyse data. In this

perspective, the reviewer is interested in the number of ultrasoundly

detected lymph nodes : only one, as generally reported in text books ?

b. Compared these parameters between confirmed cases (i.e. true plague, with

a direct demonstration of Y. pestis in any accurate clinical sample, including

buboes !) with that of excluded cases. Doing so, authors could calculate the

predictive values (both positive and negative) of this ultrasound sign, plus this

one, plus this one ….

4. To correct the following points :

a. Abstract : clarify here (and in the text) whether the study was blind or not, for

radiologists ? « Normal » morphology for buboes is a hard concept (as well as

in the text). Give topography for buboes. Clarify that only one diseased lymph

node was detected, in how many patients, in how many anatomical sites ?

b. Authors summary is definitely too long (Yersinia pestis, lien 41).

c. Line 75 : The reviewer argues that enteric form is nowadays a prevalent form ;

and that in fact, deadly septicemia could follow iny one of the three forms

(bubonic, enteric, pulmonary).

d. Line 89 : The « s » indicates that several diseased lymph nodes could be

detected in one anatomical site ibn one patient, a KEY data completely

omitted in the present report. It is possible that this parameter is of interest

for the differential ultrasoud diagnosis of buboe ; eventually usefull for the

follow-up ( ??) in treated patients. Authors HAVE to go back to data to include

this parameter in reading form and compare really confirmed patients (not

the WHO criteria, medicine !) with firmly excluded patients..

e. Table 1 adding nothing, has to be deleted.

f. Lines 191-192 : Figures unclear : how is-it possible to have 10 buboes for 12

cases, when buboes in the portal of entry ?

g. Lines 193-196 : Give figures for firmly diagnosed and firmly excluded cases.

h. Line 203 : Which treatment ? What about excluded cases ?

i. Was laterality (right / left hand using) noted ? If so, did laterality correlate

with that of buboes (especially for the upper body) ?

Reviewer #2: This is a very interesting and unique study to evaluate bubos of plague by ultrasound. The uniqueness of the study is the strength.

Minor points.

1. Ultrasound machine used in the study should be described.

2. Antibiotic treatment used in the cases should be stated.

3. Figures need improvement to appear better. I suggest changing the horizontal bar graphs comparing between day 1 and day 11 to vertical bars without Absent, making the comparison much easier.

PLOS authors have the option to publish the peer review history of their article (what does this mean? ). If published, this will include your full peer review and any attached files.

**Do you want your identity to be public for this peer review?** For information about this choice, including consent withdrawal, please see our Privacy Policy .

Reviewer #1: No

Reviewer #2: **Yes:** kentaro iwata

**Figure resubmission:**
---

## [Decision Letter · Decision Letter 1]

21 Jan 2026

PNTD-D-25-01601R1

A prospective cohort study to describe the morphology of buboes in patients with bubonic plague using ultrasound imaging

Dear Dr. Bourner,

Thank you for submitting your manuscript to PLOS Neglected Tropical Diseases. After careful consideration, we feel that it has merit but does not fully meet PLOS Neglected Tropical Diseases's publication criteria as it currently stands. Therefore, we invite you to submit a revised version of the manuscript that addresses the points raised during the review process.

Please submit your revised manuscript within by Feb 20 2026 11:59PM. If you will need more time than this to complete your revisions, please reply to this message or contact the journal office at plosntds@plos.org. Please include the following items when submitting your revised manuscript:

We look forward to receiving your revised manuscript.

Kind regards,

Joseph M. Vinetz

Section Editor

Joseph Vinetz

Section Editor

Shaden Kamhawi

co-Editor-in-Chief

Paul Brindley

co-Editor-in-Chief

**Journal Requirements:**

1) We note that your Data Availability Statement is currently as follows: "Aggregated data is available and openly accessible via the following link: https://projects.vertex.isaric.org/?param=Bubo_Images. The protocol is available on the ISARIC website: https://isaric.org.". Please confirm at this time whether or not your submission contains all raw data required to replicate the results of your study. Authors must share the “minimal data set” for their submission. PLOS defines the minimal data set to consist of the data required to replicate all study findings reported in the article, as well as related metadata and methods (https://journals.plos.org/plosone/s/data-availability#loc-minimal-data-set-definition).

2) Please amend your detailed Financial Disclosure statement. This is published with the article. It must therefore be completed in full sentences and contain the exact wording you wish to be published.

1) State the initials, alongside each funding source, of each author to receive each grant. For example: "This work was supported by the National Institutes of Health (####### to AM; ###### to CJ) and the National Science Foundation (###### to AM).".

**Reviewers' Comments:**

Reviewer's Responses to Questions

**Key Review Criteria Required for Acceptance?**

**Methods**

-Are the objectives of the study clearly articulated with a clear testable hypothesis stated?

-Is the study design appropriate to address the stated objectives?

-Is the population clearly described and appropriate for the hypothesis being tested?

-Is the sample size sufficient to ensure adequate power to address the hypothesis being tested?

-Were correct statistical analysis used to support conclusions?

-Are there concerns about ethical or regulatory requirements being met?

Reviewer #1: (No Response)

Reviewer #2: -Are the objectives of the study clearly articulated with a clear testable hypothesis stated?

Yes

-Is the study design appropriate to address the stated objectives?

Yes

-Is the population clearly described and appropriate for the hypothesis being tested?

Yes

-Is the sample size sufficient to ensure adequate power to address the hypothesis being tested?

Not relevant

-Were correct statistical analysis used to support conclusions?

Not relevant

-Are there concerns about ethical or regulatory requirements being met?

No.

**Results**

-Does the analysis presented match the analysis plan?

-Are the results clearly and completely presented?

-Are the figures (Tables, Images) of sufficient quality for clarity?

Reviewer #1: (No Response)

Reviewer #2: -Does the analysis presented match the analysis plan?

Yes

-Are the results clearly and completely presented?

UYes

-Are the figures (Tables, Images) of sufficient quality for clarity?

No. The images are blur as PDF. Should show better.

**Conclusions**

-Are the conclusions supported by the data presented?

-Are the limitations of analysis clearly described?

-Do the authors discuss how these data can be helpful to advance our understanding of the topic under study?

-Is public health relevance addressed?

Reviewer #1: (No Response)

Reviewer #2: -Are the conclusions supported by the data presented?

Yes.

-Are the limitations of analysis clearly described?

Yes

-Do the authors discuss how these data can be helpful to advance our understanding of the topic under study?

Yes

-Is public health relevance addressed?

Yes

**Editorial and Data Presentation Modifications?**

Reviewer #1: (No Response)

Reviewer #2: (No Response)

**Summary and General Comments**

Reviewer #1: The revised version PNTD-D-25-0160R1 by Josephine BOURNER and collaborators does not resolve issues highlighted by the first round review. In its present form, the report adds nothing to knowledge, regarding the natural history of plague in Madagascar; adds nothing to the diagnosis (especially differential diagnosis) of plague in Madagascar; most probably will not modify the health-care system management of plague; in Madagascar and elsewhere.

Authors remained blind to the basic remarks of the reviewer, correcting only a few minor details.

At a glance:

1- The absence of any control group, either apparently healthy matched people / either patients with other acute infection, is just rendering impossible to reach goals of the study.

2- The on-going affirmation of “normal buboes” is just non-sense.

3- The still discrepancy in data, illustrated by contradictory line 20 / line 29 in the abstract.

4- The absence of profound lymph nodes ultrasonographic examination, where ultrasonography would be most valuable: in fact, in the study, clinics (palpation n = 12) was superior to ultrasonography ( n= 10)!

Reviewer #2: Well written descriptive study. The images are blur as PDF downloaded for the reviewer. Should appear better.

PLOS authors have the option to publish the peer review history of their article (what does this mean? ). If published, this will include your full peer review and any attached files.

**Do you want your identity to be public for this peer review?** For information about this choice, including consent withdrawal, please see our Privacy Policy .

Reviewer #1: No

Reviewer #2: **Yes:** Kentaro Iwata

**Figure resubmission:**
---

## [Editor Report · Decision Letter 2]

9 Mar 2026

Dear Ms Bourner,

We are pleased to inform you that your manuscript 'A prospective cohort study to describe the morphology of buboes in patients with bubonic plague using ultrasound imaging' has been provisionally accepted for publication in PLOS Neglected Tropical Diseases.

Best regards,

Joseph M. Vinetz

Section Editor

Joseph Vinetz

Section Editor

Shaden Kamhawi

co-Editor-in-Chief

Paul Brindley

co-Editor-in-Chief

---

## [Editor Report · Acceptance letter]

Dear Ms Bourner,

We are delighted to inform you that your manuscript, "A prospective cohort study to describe the morphology of buboes in patients with bubonic plague using ultrasound imaging," has been formally accepted for publication in PLOS Neglected Tropical Diseases.

Best regards,

Shaden Kamhawi

co-Editor-in-Chief

Paul Brindley

co-Editor-in-Chief
